# Interpersonal coordination analysis in bat-and-ball sports under a real game situation: Asymmetric interaction and delayed coupling

**Ryota Takamido** [1]*, **Keiko Yokoyama**[2], **Hiroki Nakamoto**[3], **Jun Ota**[1], **Yuji Yamamoto**[2]

**1** Research into Artifacts, Center for Engineering (RACE), School of Engineering, The University of Tokyo, Tokyo, Japan, **2** Research Center of Health, Physical Fitness & Sports, Nagoya University, Nagoya, Japan, **3** Faculty of Physical Education, National Institute of Fitness and Sports in Kanoya, Kanoya, Kanoya City, Kagoshima, Japan

* takamido@race.t.u-tokyo.ac.jp

**Data Availability Statement:** All relevant data are within the paper and its Supporting Information files.

## Abstract

This study investigated the interpersonal coordination between the pitcher and the batter in bat-and-ball sports. Although the importance of interpersonal coordination is widely accepted in many sports, no studies have investigated it in bat-and-ball sports because the dominant task constraints surrounding the interaction between pitcher and batter make it difficult to apply conventional analytic techniques. To address the issue, this study proposes a new analytical framework to investigate interpersonal coordination in bat-and-ball sports under a real game situation with two main characteristics: asymmetric interaction and delayed coupling. First, the dynamic time warping technique was used to evaluate the stability of the head movement pattern of the pitcher and batter, and cross-correlation analysis was used to quantify the temporal relationship between them. We found that the head movement pattern of batters was significantly more unstable than that of pitchers, and approximately 60% of the variance of the change in the head movement pattern of batters could be explained by that of the pitchers. Moreover, expert batters followed a pitcher's movements with a specific time delay of approximately 250 ms. These findings highlight the characteristics of interpersonal coordination in bat-and-ball sports: the pitcher can make a pre-patterned stable motion, whereas the batter needs to follow and adjust their movement to it. Although the effects of prediction ability need to be investigated to understand its detailed mechanism, the contribution of this study is that it revealed the existence of the interpersonal coordination between the pitcher and batter of bat-and-ball sports under a real game situation.

## 1. Introduction

Interacting with other players is a critical factor in many sports. For example, if football players can accurately understand teammates' intentions and pass the ball to the right place at the right time, the pass success rate significantly increases. Previous studies have shown that

**Funding:** This work was supported by Japan Society for the Promotion of Science (Grant number: JP22K17712). The funders had no role in study design, data collection and analysis, decision to publish, or preparation of the manuscript.

**Competing interests:** The authors have declared that no competing interests exist.

different sports require different types of interpersonal coordination because of the need to adapt to the dominant task constraints of each sport, as seen in sports such as football [1, 2], rugby [3], basketball [4], tennis [5, 6], badminton [7], kendo [8, 9], boxing [10], and Aikido [11]. In other words, how one interacts with other players is highly dependent on how one's movements need to conform to external parameters such as sporting rules, tools, or playing area. Therefore, analysis of interpersonal coordination styles across different sports can bring us different insights into human coordination in the athletic context.

However, in the case of bat-and-ball sports, such as baseball, cricket, and softball, researchers have mostly focused on single players among pitchers (defenders) and batters (attackers), and no studies to date have analyzed the relationship between them, although the results of interviews with experts suggest its importance [12]. This is likely because the two dominant task constraints that affect the relationship between pitcher and batter are different from those of other sports, thus making it difficult to use conventional analytical techniques. Fig 1 shows the schematic image of two characteristics in the interpersonal coordination in bat-and-ball sports. Although these characteristics are partially observed in other sports, such as soccer penalty kicks and tennis services, these relations can be maintained throughout the entire game in bat-and-ball sports.

First, in contrast to other sports, one player (the pitcher) always takes the initiative, while the other player (the batter) needs to observe the pitching movement and respond to it; this is called an asymmetric interaction (Fig 1A). In other sports, although the attacker holds a dominant position, both offenders and defenders must observe their opponents' actions and adjust their movements accordingly. However, in bat-and-ball sports, pitchers start and complete the pitching movements at their own pace and timing without being blocked by the batter; hence, they can perform pre-patterned and self-paced pitching movements, concentrating on making fast and accurate pitches. Batters, in turn, need to observe the pitch to predict the attributes of the ball, such as type of throw (e.g., fastball or curveball) and projected location of the ball at the plate, and perform preparatory movements on the basis of these observations. They cannot initiate or complete the hitting movement until the pitcher's movement is already in progress.

Additionally, given that hitting movements should be performed toward the pitched ball and not the pitcher, there is a time delay between the batter's and pitcher's actions that takes into account the batter's prediction and preparation for the pitched ball; this temporal relationship is called delayed coupling (Fig 1B). Other sports require an immediate response to an opponent's actions such that a delayed response leads to failure; however, in bat-and-ball sports, a delay is necessary. The batter must observe the pitching movement, predict when the ball will arrive, and intersperse the appropriate delay on the basis of this movement. Although a batter's ability to predict the movement of the pitched ball and perform adjustments on their hitting movement pattern have been investigated [13–19] and given that the importance of these skills for successfully hitting the ball has been widely accepted [20], previous studies have mostly focused on the performance of a single player (the batter), and no study has investigated the temporal relationship between pitcher and batter organized by a batter's prediction ability. Some studies have measured the relative phase between two players to evaluate the degree of movement synchrony in other sports (e.g., Duarte, et al. [1]; Travassos, et al. [21]). However, given that both pitching and hitting movements are acyclic and discontinuous (i.e., both pitchers and batters move forward at once for each pitch), it is difficult to apply the concept of relative phase to bat-and-ball sports.

In light of this background, in this study, we analyzed aspects of interpersonal coordination in bat-and-ball sports in a real game situation while focusing on these two characteristics (Fig 1). Specifically, we analyzed the head movement patterns of both pitchers and batters on the basis of the recorded images of real softball games. In bat-and-ball sports, both pitchers

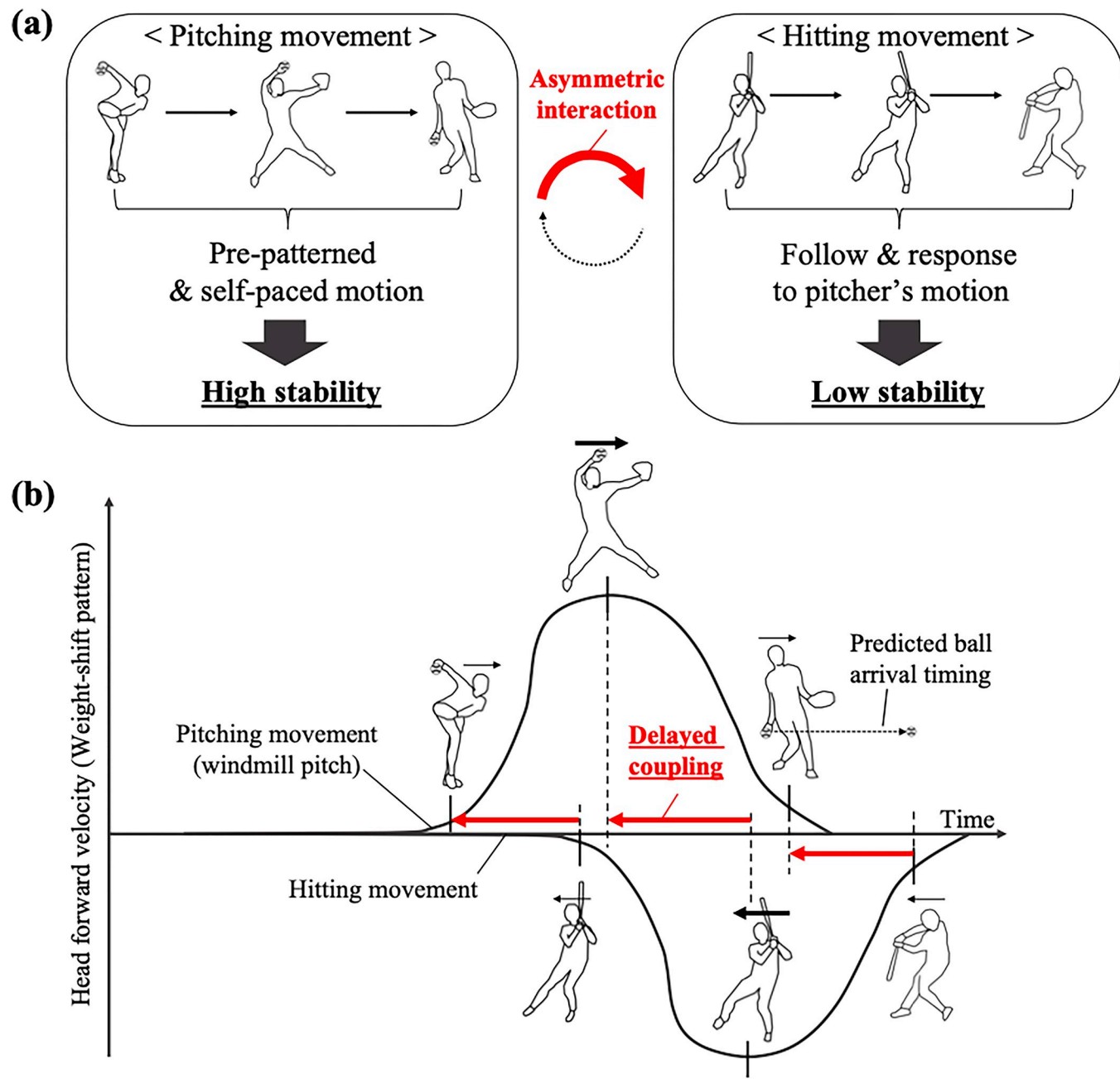

**Fig 1.** The characteristics of interpersonal coordination in bat-and-ball sports: (a) asymmetric relationship between a pitcher and a batter and (b) delayed coupling in which the batter follows the pitcher's movements.

and batters shift their weights forward and backward to generate kinetic energy, and this weight-shift behavior is often reflected in head movement (although not always). In addition to head movement, there is a possibility that information on other body components is used for the interaction. Nevertheless, "global body positioning" [20] is the primary task in the preparation (coordination) phase before a pitcher's ball is released. Additionally, a previous study has reported that a batter's adjustment ability in response to the difference in a pitcher's kinematic information [18], as well as forward and backward directional head movements, could

be clearly observed even for skilled batters who sometimes has the advice not to move their head in the swing phase [22]. Therefore, we consider it is an appropriate index for the analysis of the interpersonal coordination in bat-and-ball sports.

On the basis of the two characteristics shown in Fig 1, we set two hypotheses for the analysis. First, it can be assumed that, if the batter's movement is more affected by the pitching movement (Fig 1A), the head movement pattern of the batter is more unstable across different pitches than is that of the pitcher, and the changes in the batter's movements can be explained by changes in the pitcher's movement (Hypothesis 1). In addition, because batters must follow the pitcher's movement while predicting the pitched ball (Fig 1B), there is a consistent time delay between them that is close to the ball travel time (Hypothesis 2). Through these analyses, we sought to clarify the nature of interpersonal coordination in bat-and-ball sports is a way that is sensitive to the dominant task constraints of the sport, obtaining novel insights into interpersonal dynamics within an authentic gaming context.

## 2. Materials and methods

### 2.1. Data for analysis

We used data collected in a previous study that recorded both batter and pitcher movements during a real softball game [23]. Softball is a bat-and-ball sport in which the pitcher throws the ball according to a specific pitching method called "windmill," and the batter tries to hit the ball using hitting movements similar to those in baseball [24]. Although real-life recordings may provide less accurate data compared to motion capture or other measurement systems, it does afford gestalt-level analysis of authentic interpersonal coordination as the first step in exploring this novel research area.

This study was approved by the university's Institutional Review Board for Human Subjects Research (ethical number: 28–02) during the academic year 2016–18 and conformed to the principles of the Declaration of Helsinki. Written informed consent was obtained from all participants of this study. The data used in this study was collected from 16 official softball games from the 2016 academic year. The authors did not access to information that could identify individual participants after data collection. The individual pictured in Fig 2 has provided written informed consent (as outlined in PLOS consent form) to publish their image alongside the manuscript.

An overview of the data collection is provided in Fig 2. The pitcher and batter's movements were recorded during an official softball game by two video cameras at 60 frames-per-sec (fps) from the side angle; the information used for analysis was the time series of image information of the pitcher and batter under a real game situation. Two video cameras were optically synchronized on the basis of the information of the light bulbs flashing in the same viewing field. Considering that we focused on the interaction between the pitcher's pitching movement and the batter's hitting movement before ball release, we included data where the batter did not make a swing movement for the analysis. Given the limitation of the structure of the ballpark, we sometimes need to slightly change the viewing angle from the side angle (Fig 2); however, given that the amount of head movement of players large enough to discriminate the direction and because we normalized the head movement data, we considered that it had the acceptable accuracy for the analysis for this study.

The participants were 45 batters and 12 pitchers from the Japan women's softball league (JSL). The league consists of company teams and has the highest competition level in Japan; most players of the national softball team play in this league. The inclusion criterion was players who have pitched more than nine balls in one game. This criterion was set on the basis of the following: the batter is given a minimum of three pitches (three strike counts) for

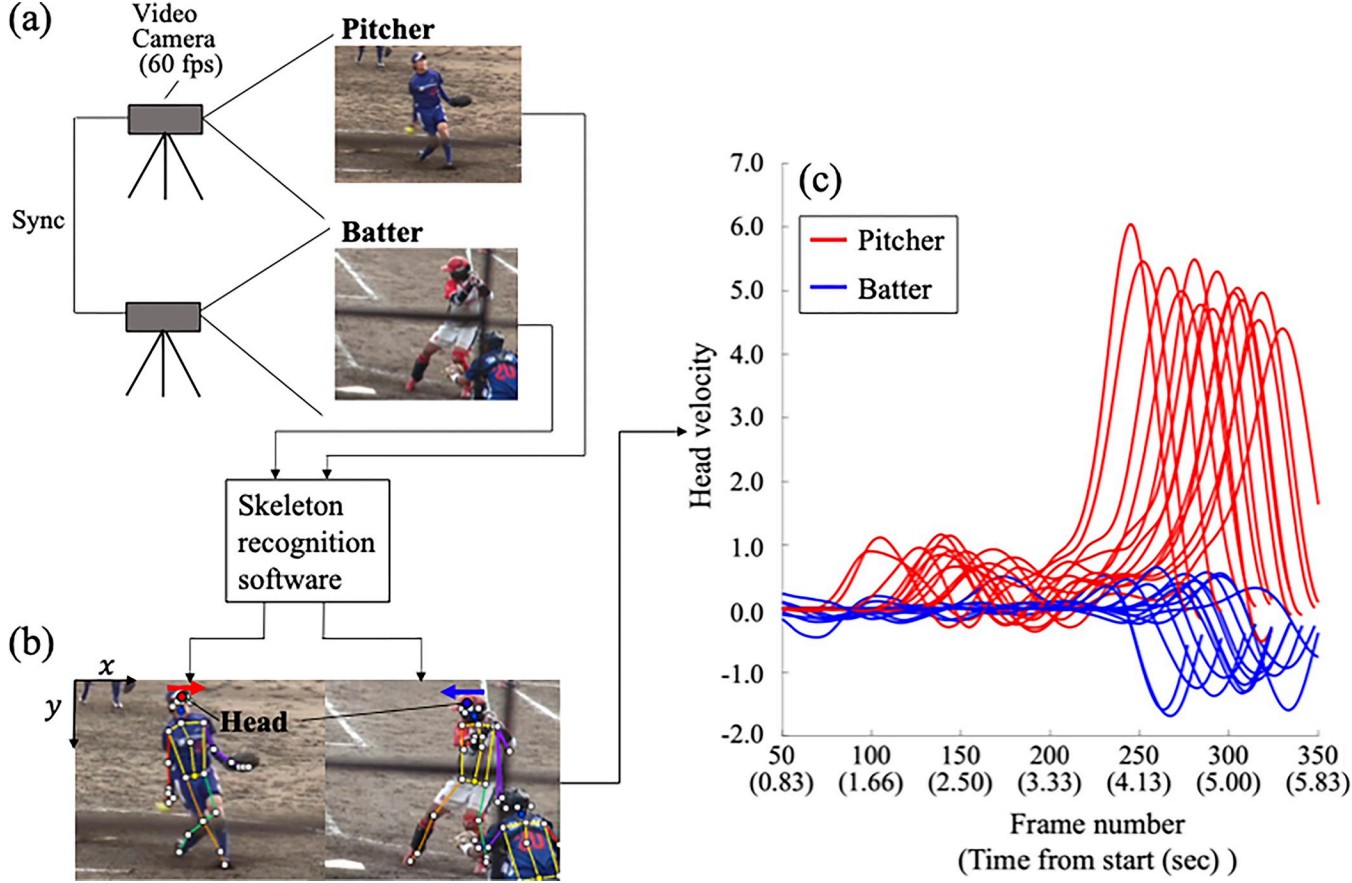

**Fig 2. Data process used in this study.** (a) Example of the image data in a real-world softball game. (b) Obtained images from the skeleton recognition software. (c) Head velocity data of one pair of pitcher and batter calculated by the process (12 pitches).

constructing the relationship between the pitcher in an at-bat; three at-bats are given to a batter in a game; and nine balls (three pitches and three at-bats) can be set as a criterion for the analysis of interpersonal coordination in real game contexts. However, there were duplications of the pitcher such that different batters played against the same pitcher. Considering that we focused on the interaction between pitchers' and batters' movements, we set additional criteria for data selection; if there was a duplication of the pitcher among the 45 league batters, we selected the batter with the highest number of pitches (trials). As a result, we gathered data from 12 pairs of expert batters and pitchers, with more than 9 years experiences of softball, for the analysis, with a mean number of pitches of 12.0 ± 2.6.

On the basis of the image data during the softball game, we extracted information on the head movement in the forward direction of both the pitcher and batter. We used the skeleton recognition software VisionPose (NEXTSYSTEM Co., Ltd., Fukuoka, Japan) to extract head position information (Fig 2B). The software automatically recognizes human skeletal information in the images and returns two-dimensional position information (pixel values for x and y) of each joint. We measured the forward directional (x-coordinate) head position. In cases when skeleton recognition was unstable owing to the inclusion of other persons in the image (e.g., referees and field players), we manually recorded head position with a MATLAB (The MathWorks, Inc., Natick, MA, USA) program that returned the clicked coordinate point in the images. The head position data of the pitcher and batter were smoothed using a fourth-

order Butterworth filter with a cut-off frequency of 6 Hz. The data were then normalized with Z-scores to exclude the effect of differences in the camera's magnification ratio across games. Head velocity was then calculated using the time derivative of the head position data. Fig 2C shows an example of the head velocity data of one pitcher-batter dyad calculated using these processes. Given that the directions of head movement of the pitcher and batter are different, they have peaks on opposite sides. The time range of the data for one pitch is set from when the pitcher demonstrates the set-up pose (more than a 2-sec pause is required by the official softball rules) to when the ball arrives at the contact point. As shown in Fig 2C, we mostly focused on the interaction between the pitcher and batter before ball release rather than during the ball-flight phase; this approach is different from those of previous studies, which mainly focused on the swing-execution phase (e.g., [25]). In cases when the batter chose not to swing, the ball arrival time was defined as the time when the ball crossed the front edge of home base.

## 2.2. Analysis

### 2.2.1. Analysis 1: Evaluation of head movement pattern stability and dependency by the dynamic time warping technique.

To verify Hypothesis 1, we used the dynamic time warping (DTW) technique to evaluate the head movement pattern stability of each expert pitcher and batter as well as the dependency of the batter's head movement on that of the pitcher. DTW is an analytic technique used to calculate the optimal alignment and similarity between two time-series datasets [26]. As it can calculate the similarity between two time-series data of different lengths, it is used to distinguish and recognize human movement patterns [27, 28].

Fig 3 shows a schematic diagram of the DTW process. Given two time-series data, namely, $Q = q_1, q_2, . . ., q_n$ and $C = c_1, c_2, . . ., c_m$, and the $m \times n$ distance matrix of $W = w_{11}, . . ., w_{nm}$, the optimal path from $w_{11}$ to $w_{nm}$ was calculated to minimize the pass among them in a grid field; this calculation returns the alignment indexes $I = i_1, . . ., i_k$ and $J = j_1, . . ., j_k$ for each time-series datum of the same length. The aligned time-series data are represented by $Q\prime = q_{i_1}, q_{i_2}, . . ., q_{i_k}$ and $C\prime = c_{j1}, c_{j2}, . . ., c_{jk}$. By referring to the indexes of the alignment data, we can confirm the temporal relation between the two time-series data (e.g., $q_1$ corresponds to $c_3$).

In this study, we applied DTW to Euclidean distances from two perspectives. First, to evaluate the stability of the head movement pattern of each pitcher and batter, the similarity in the shape of the head velocity data of each pitcher and batter across different pitches was evaluated by DTW (Fig 4A). If the batter's movement was affected more by the pitcher's than vice versa, as in Hypothesis 1, their head movement pattern would be more unstable and show lower

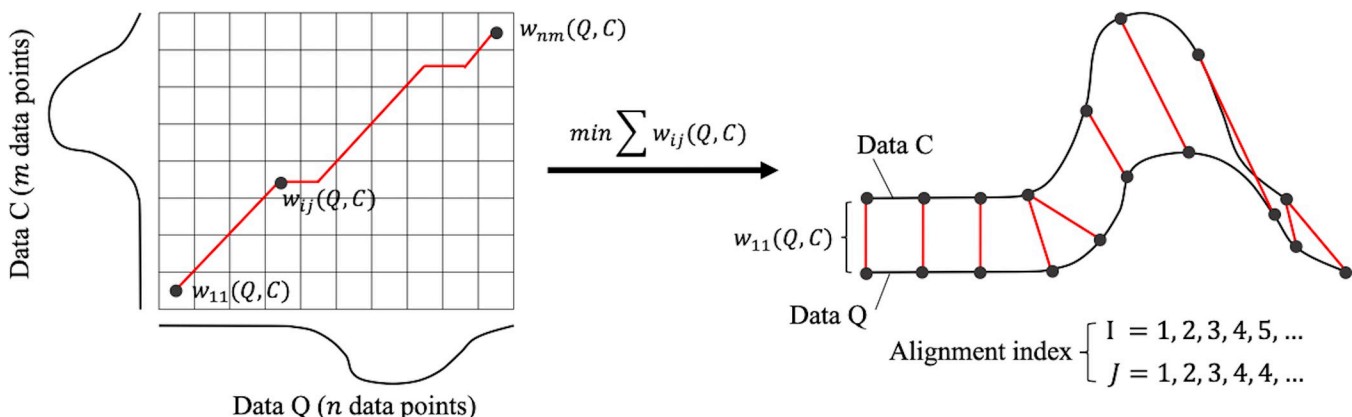

**Fig 3. Schematic diagrams of the DTW technique.**

similarity than that of the pitcher. Hence, the distance calculated by DTW becomes larger, and the aligned time-series data are more unmatched in batters than in pitchers. Specifically, we calculated the mean distance between head velocity data with different pitches for each of the 12 pairs of batters and pitchers, and then calculated the mean value. Because DTW is a one-to-one comparison method between two time-series datasets, we repeatedly applied it to all combinations of pitches when calculating the mean value for each player. For example, if the total number of pitches was 10, we applied DTW 45 times and calculated the mean distance between them (i.e., 1st pitch vs 2nd pitch, 1st pitch vs 3rd pitch, etc.). In addition, to consider the differences in data points and the min-max range of the head velocity data, the distance value was divided by the number of data points and the value of the subtraction of the min-max values. We also calculated the $R^2$ value between two aligned time-series data obtained as a result of DTW and calculated the mean $R^2$ value for the expert batters and pitchers.

In the statistical analysis, to confirm whether the stability of the movement patterns of pitchers and batters was significantly different, we applied a non-paired $t$-test to the mean distance and $R^2$ values of batters and pitchers. The significance level was set at 5% for each statistical analysis, and the 95% confidence interval (95% CI) for the mean difference between the two groups was calculated using $t$-tests. To confirm the effect of the sample size of this study, the effect size (Cohen's d) was calculated for $t$-tests, and we also performed a power analysis to calculate the smallest possible effect size with an alpha error probability of 0.05 and statistical power of 0.80. The results revealed the smallest possible effect size that could be detected with our sample (12 pitchers and 12 batters) was 1.19. The $t$-test was performed using R Statistical Software (R Development Core Team, Vienna, Austria), and calculation of the effect size and power analysis were performed with G*Power Software version 3.1 [29].

Additionally, we calculated the dependency of the head movement of the batter on the pitcher by using the DTW technique (Fig 4B). If the differences in a batter's movement across different pitches were caused by the response to the differences in pitching movement, there would be a common temporal relationship (alignment indices) between them, which would explain the changes in the hitting movement due to changes in pitching movement. Fig 4B shows a schematic diagram of the analysis. On the basis of the alignment index data of the pitching movement between different pitches, namely, $i_{pn}$ and $i_{pm}$, which can be obtained from the above analysis, the batter's head velocity data of the correspondent pitches were shifted by an amount equivalent to those indexes. The new batter's movement data were then calculated as $B_n(i_{pn})$ and $B_m(i_{pm})$, which is aligned with the information of pitcher's movement data. The evaluation of the dependency is performed by calculating the R-squared value between $B_n(i_{pn})$ and $B_m(i_{pm})$ for each pair of batter and pitcher.

**2.2.2. Analysis 2: Quantification of the time delay between pitcher's and batter's movements.** To verify Hypothesis 2, we quantified the time delay in the head movement of the pitcher and batter using a cross-correlation analysis. Cross-correlation analysis is a technique used to quantify the concurrency of two time-series data. Although it has not been used for performance analysis in the field of sports science, it is widely used to identify the temporal structure between time-series data in other research domains, such as identifying neuron networks [30]. Given that it can be applied to nonperiodic motion with no data length restrictions, we introduced this method to clarify the temporal structure between players in bat-and-ball sports. Fig 5 shows a schematic diagram of the analysis. Given two time-series datasets with length, the cross-correlation function is represented by the following equation:

$$C_{XY}(\tau) = \sum_{t=1}^{T} X(t)Y(t + \tau). \tag{1}$$

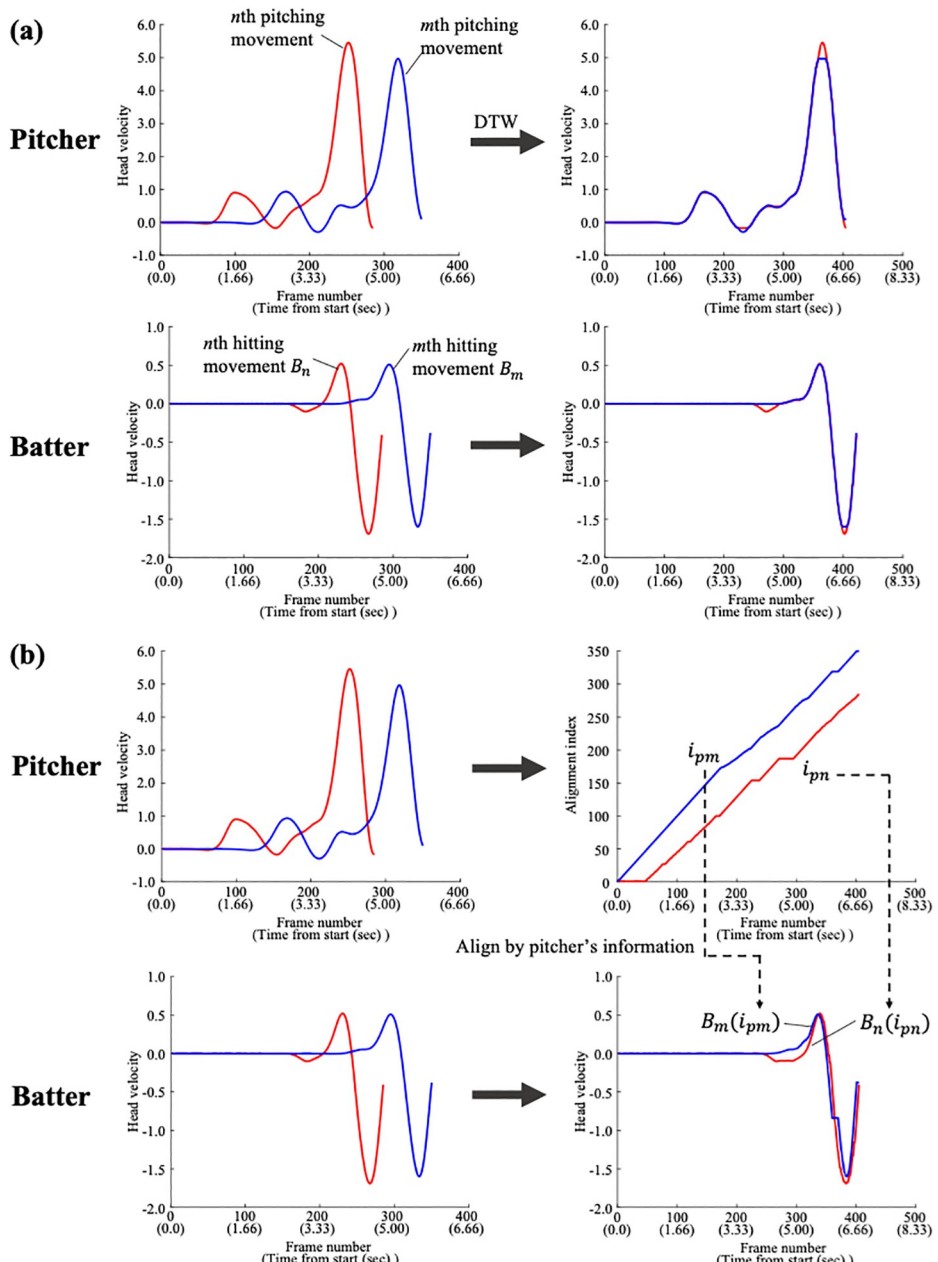

**Fig 4. DTW analysis of the head movement patterns of the pitcher and batter.** (a) Evaluation of the movement pattern stability of each pitcher and batter. (b) Calculating dependency of the batter's movement on the pitcher's movement. The values in brackets on the horizontal axis indicate the actual time.

In this study, we calculated the optimal time delay, which maximizes the cross-correlation between the pitcher's and batter's head velocity data in the same pitch. If the batter immediately responds to the pitcher's movement, the cross-correlation function is maximized at $\tau \cong 0$. If the batter responses pitcher's movement with a time delay, the cross-correlation function is maximized at $\tau > 0$, and it is assumed to be close to the mean ball travel time of national league players (0.41 sec or ~24 frames) [23]. In statistical analysis, we calculated the mean optimal time delay for each pairing of batter and pitcher and then calculated the mean value among all

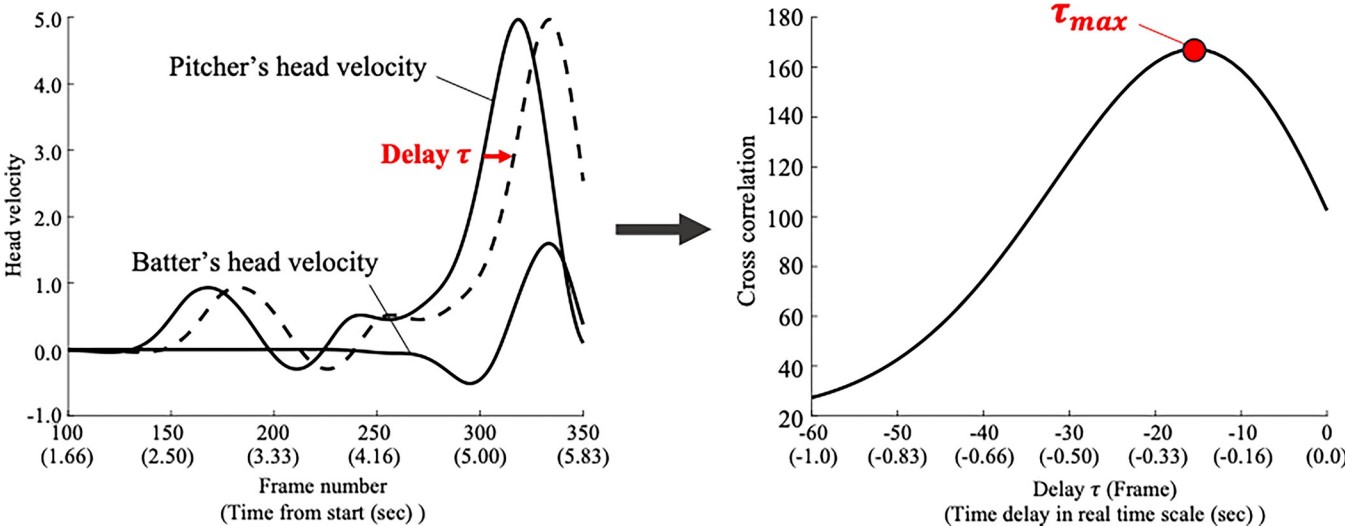

**Fig 5. Analysis of the temporal delay between the pitcher's and batter's head movement.** The values in brackets on the horizontal axis indicate the actual time.

pairs. When we calculated the cross-correlation to adjust for the differences in the head movement direction in the image, the batter's data were multiplied by minus one. To verify whether the hitting movement of the batter couples pitching movement of the pitcher with temporal delay, one-sample t-test with $\tau = 0$ as the criteria was applied to mean value. We also calculated the correlation coefficient between the pitcher's head velocity data and the batter's head velocity data shifted by $\tau_{max}$. The significance level was set at 5%, and the effect size was calculated. From the results of the power analysis, the smallest possible effect size for the one-sample t-test and given sample size was calculated as 0.89.

## 3. Results

### 3.1. Analysis 1

Fig 6 shows the results of Analysis 1 for the evaluation of the movement pattern stability of each pitcher and batter. As shown in Fig 6, congruent with our hypothesis, the movement pattern of the pitcher was more stable and consistent across pitches than that of the batters. The t-test results revealed that, although both pitchers and batters showed high consistency of their movement patterns (mean $R^2$: $0.99 \pm 1.0 \times 10^{-3}$, for pitchers and $0.92 \pm 5.0 \times 10^{-2}$ for batters), the $R^2$ values of the aligned time-series data across different pitches were significantly larger in pitchers than batters, $t(22) = 4.7$, $p < .01$, 95% CI ($3.5 \times 10^{-2}$, $9.2 \times 10^{-2}$), and the difference had enough statistical power (d = 1.90 > 1.19). Moreover, the mean distance between two time-series datasets calculated by DTW was smaller in pitchers than batters, $t(22) = 3.1$, $p < .01$, 95% CI ($2.07 \times 10^{-2}$, $4.2 \times 10^{-2}$), d = 1.59 > 1.19. These results suggest that, in bat-and-ball sports, while the pitcher iterates a pre-patterned pitching movement, the batter changes their hitting movement pattern more significantly between pitches.

In addition, Fig 7 shows the $R^2$ value of the batter's head velocity data aligned by the DTW indices for the pitching movement data of different pitches. Consistent with our hypothesis, most pairs of batters and pitchers showed a strong relationship between their head movement patterns. On average, over 60% of the variance in the batter's head movement pattern across different pitches could be explained by the information on changes in the pitching movement

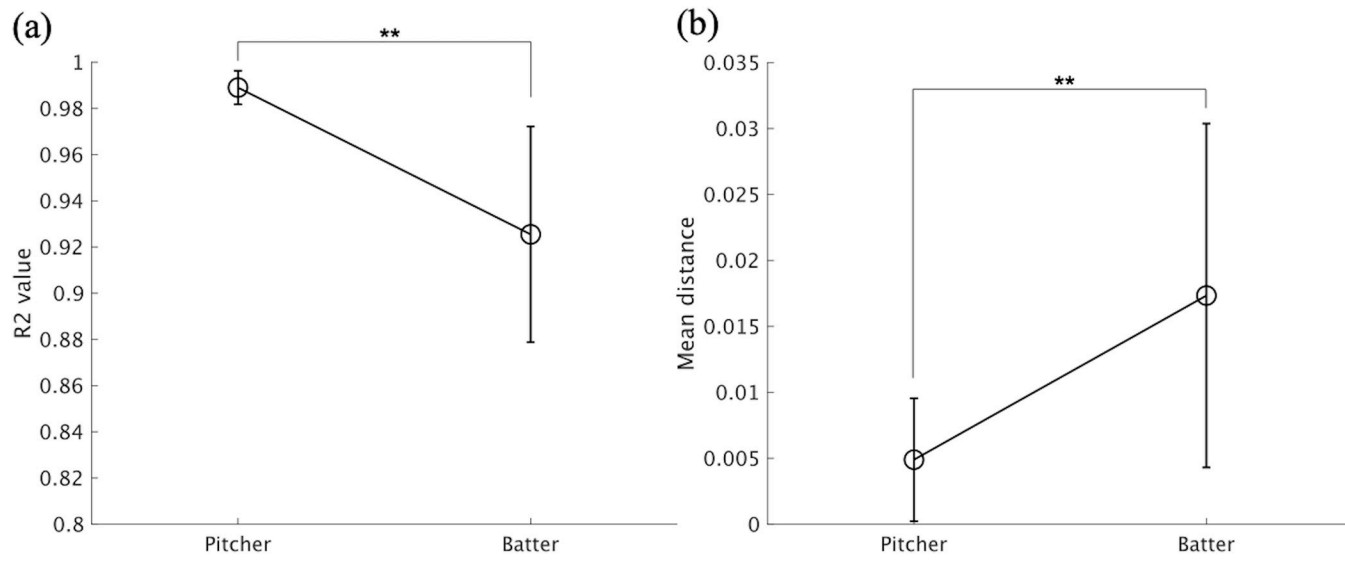

**Fig 6. Movement pattern stability of the batter and pitcher in a softball game.** Error bars represent the standard deviations among 12 pitchers and batters. ** $p < .01$.

pattern (mean $R^2$ value: 0.63 ± 0.13). These results suggest that batters change their hitting movement patterns in response to changes in their opponents' pitching movement patterns.

### 3.2. Analysis 2

Fig 8 presents the results of Analysis 2. The cross-correlation was maximized at $\tau > 0$ in all pairs. The mean optimal time delay $\tau_{max}$ was 0.25 ± 0.10 sec, and one-sample $t$-tests revealed a significant time delay between the pitchers' and batters' head movement patterns, $t(22) = 8.8$, $p < .01$, 95% CI (0.19, 0.31), d = 2.50 > 0.89. The absolute correlation coefficients of the

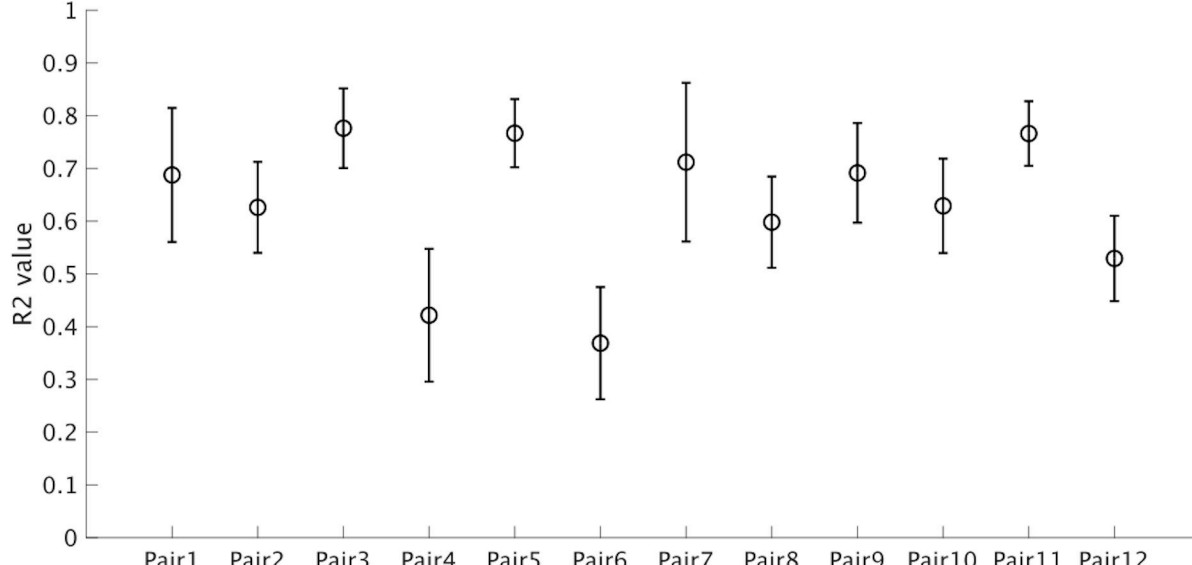

**Fig 7. Dependency of the differences of hitting movement patterns on pitching movement patterns.** Error bars represent the standard deviations for each pair of pitcher and batter.

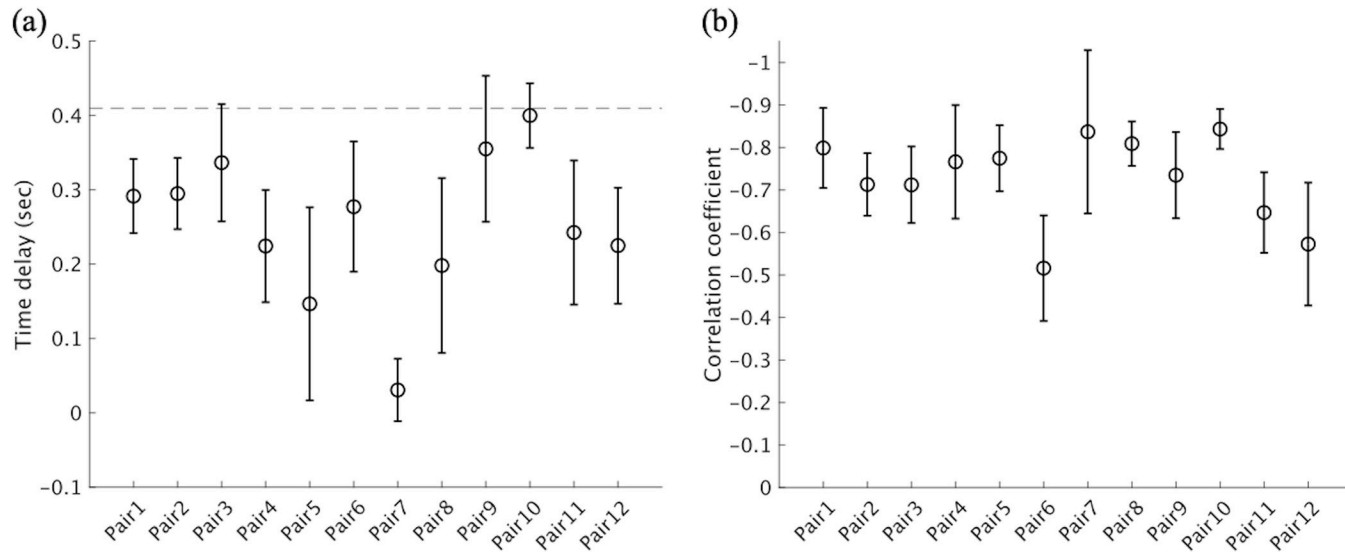

**Fig 8. Results of the analysis for the time delay between the pitcher's and batter's movements.** (a) Mean time delay of the batter's movement of all pairs. (b) Mean correlation coefficient between a pitcher's and batter's head velocity data with time shift. Error bars indicate standard deviations among the 12 players. The dotted line indicates the mean ball travel time of national league pitchers (0.41 sec).

pitching and shifted hitting movement data for all pairs were more than 0.5. Fig 9A shows an example of the temporal relationship between the pitcher and batter with a velocity–velocity plot for the same pitch for one pair (12 pitches of Pair 1). Fig 9B shows an example of the relationship between the pitcher's head velocity data and the batter's head velocity data shifted by the amount of optimal time delay. As seen in Figs 8 and 9, these results partially support Hypothesis 2; that is, expert batters do not react to the pitcher's movement immediately; instead, their own movements follow with a certain time delay, which is smaller than the ball

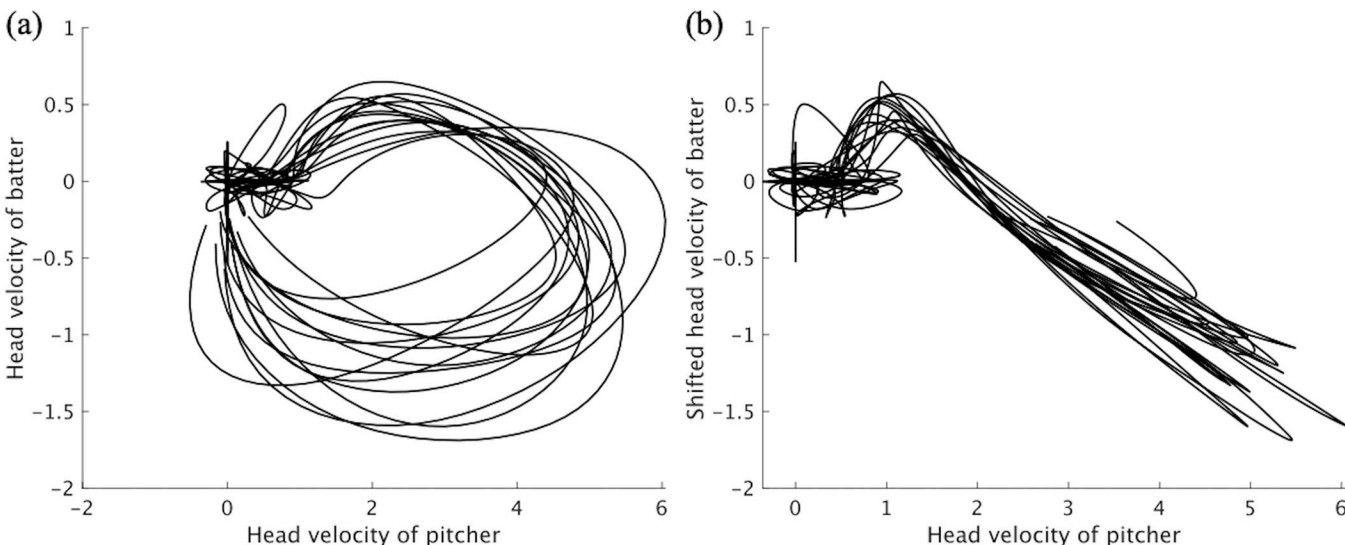

**Fig 9. Example of the temporal relationship between the pitcher's and batter's head velocity data (Pair 1).** (a) Velocity–velocity plot with head velocity data of the pitcher and batter. (b) Velocity–velocity plot with head velocity data of the pitcher and that of the batter's shifted data. Note that the value of each axis is a dimensionless quantity and that the absolute value of the number has no physical meaning.

travel time of 0.41 sec. Rather, batters seemed to adjust the temporal delay to avoid exceeding the ball travel time (Fig 8).

## 4. Discussion

In this study, we investigated and analyzed the head movement patterns of the pitcher and batter in a real softball game situation and obtained novel insights into their interpersonal coordination. First, the results of DTW showed that the movement pattern of the pitcher was more stable among different pitches than that of the batter (Fig 6). As mentioned earlier, the specific relationship between the pitcher and batter results from the adaptation to two dominant task constraints in bat-and-ball sports. Previous studies have reported that expert athletes obtain task-dependent skills by adapting to the task constraints of each sport, such as specific rules, tools, or environments (e.g., size of playing area) [30, 31], which also affect and organize the specific relationship between players in each sport [32]. From this perspective, it seems that two dominant task constraints in bat-and-ball sports (Fig 1) organize the relationship between pitchers and batters. More specifically, because pitchers do not have to change their movement in response to the batters, they learn repetitive and accurate pitching movements. Batters, in turn, need to adjust their hitting movement in response to it.

Furthermore, the results of the calculation of the $R^2$ value with the alignment index showed that the 60% of the differences in hitting movement can be explained by the differences in pitching movement (Fig 7). However, there are individual differences among expert players in the value. Assuming the pitcher performs stable and repetitive pitching movement, this may suggest that expert batters may have their own strategy for interacting with pitchers. Although a higher $R^2$ value represents a strategy wherein batters strictly couple their movement with the pitching movement, a lower $R^2$ value indicates that batters perform more robust hitting movements on the basis of the changes in the pitching movement pattern. Hence, these discussions bring us the following important query: "Who are the experts in interpersonal cooperation? The player who can flexibly adjust their movement to others, or a player who can make robust movement regardless of others." Given that it is difficult to decide which of these two strategies is better without a detailed comparison of actual performance results, further investigations are expected in the future. Overall, as shown in these results and discussions, the DTW technique can yield knowledge of the interaction between two players, which cannot be obtained by conventional analytical techniques. Hence, this study verified the effectiveness of the DTW technique for analyzing interpersonal coordination in game contexts.

Additionally, the results of the cross-correlation analysis (Analysis 2) showed that there was a time delay between the two time-series datasets of the head velocities of the pitcher and batter. Hence, it can be considered that the relationship between pitchers and batters is closer to a "leader–follower" relationship (e.g., Meerhoff & De Poel [33]), in which one player always takes the initiative and the other always follows, than to a "defender–attacker" relationship (e.g., Passos et al. [3]), in which initiative switches between the two players even though the attacker tends to be in the dominant position. As mentioned in the Introduction, the reason for this seems to be that they interact through the pitched ball (Fig 1B). However, in contrast to Hypothesis 2, the mean time delay between the batter's and pitcher's movements was approximately 250 ms, which is smaller than the mean ball travel time of 410 ms. There are several potential reasons for this. First, to ensure a margin of time for making decisions or movement executions (approximately 160–200 ms [18]), the batter completes the head positioning earlier than the shortest time predicted for the ball arrival (i.e., the fastball case). This corresponds to the "sitting on a fastball" strategy [34], in which the batter performs the preparatory movement while assuming a fastball. From another point of view, the batter may be

"entrained" to the pitcher's movement. According to a dynamic systems approach, it is more difficult for the batter to create a specific time delay from the pitcher's movement than to synchronize with it; therefore, the batter is unconsciously entrained to the pitcher's movements [35]. Furthermore, there is a possibility that this is caused by differences in the asymmetric movement kinematics of the pitching movement (windmill pitch) and the hitting movement. If the batter requires more time than the pitcher's arm-swinging movement to complete the batting movement after completing the head movement motion, then the initiation of the batting movement must start relatively early. Finally, as shown in the next paragraph, the batter couples other body component, such as the hand to the pitcher's motion, and the delay may be matched to the ball travel time on the parts.

Finally, we consider that it is quite important to discuss the effects of the batter's prediction ability. As shown in many previous studies [20], the batters are passively entrained to the pitching movement, actively picks up the pitcher's kinematic information, predicts the trajectory of the pitched ball, and adjust their hitting movement on the basis of this movement. From this perspective, each insight obtained from the results of this study may be related to such a perceptual motor control mechanism of the batter. For example, there is a possibility that the larger variance of the hitting movement pattern of the batter is derived from the variance of the predicted ball trajectories and not only from the follow-up behavior to the pitcher because the batter can change and adjust their hitting movement depending on this pattern [15, 17, 19, 36]. Furthermore, for the pitcher, they may try to decrease the variation of the pitching motion, and this behavior can provide a clue to the batter. Additionally, it also suggests that there may be other types of interaction than the head movement. If the clue of the pitched ball trajectory appeared in the local body component information, such as the kinetics of the throwing hand or elbow, the hitting movement of the batter can be coupled with such a specific information.

Therefore, to obtain deeper insights of the interpersonal coordination in bat-and-ball sports, we also need to consider the perceptual motor control mechanism and not jus unconscious entrainment. In this context, a schematic model proposed by Müller and Abernethy [20] may be helpful for integrating these two different aspects: unconscious entrainment and active prediction. From their model, the batter first performs "global body positioning" movement on the basis of the pitcher's kinematic information at an earlier stage such as during weight shift before ball release. Considering that the batters do not have enough information for accurate prediction in the earlier phase, they passively follow the pitcher's motion while waiting to obtain further information even though some contextual information of ball delivery [37] can be used. Thereafter, the batters perform "precise spatiotemporal interception" on the basis of the latter and obtains more relevant information such as ball-flight information; this movement can be significantly affected by the prediction rather than the unconscious entrainment of the pitcher's motion. Therefore, both aspects should be investigated to understand the detailed mechanism of the interpersonal coordination in bat-and-ball sports. Obviously, this study analyzed and focused on the former interaction; hence, further investigations are required to analyze the latter type of interaction.

To summarize, it seems that bat-and-ball sports necessitate aspects of interpersonal coordination that are organized around adaptation to dominant task constraints. Although further investigations are required to understand its detailed mechanisms, including the effect of the prediction, this study revealed the existence of the coordination relationship between the pitcher and batter in real game contexts of bat-and-ball sports.

This study has some limitations to consider. First, because the materials for data analysis were two-dimensional video camera images, there might have been slight movements and positional changes that were undetected. Furthermore, we did not evaluate the relationship

between the performance of the pitcher or batter and their interactions. If a batter fails to execute the proper time delay vis-à-vis the pitcher's movements, the hitting success rate decreases; we did not consider success rates in our analysis because of the limitation of the number of successful hitting in real game context. Hence, this should be verified by conducting further experiments on the adjustment of the difficulty of the task to obtain a sufficient amount of both success and failure data. From this aspects, a virtual reality system [14, 19, 36, 37] that can easily change the difficulty of the hitting can be a potential tool. Finally, because we only investigated the interpersonal coordination of national league players, assessment of novice and intermediate players is required. The relationship between the pitcher and batter may change depending on their expertise levels. Having this knowledge could prove invaluable for coaching and skill-building training for individual athletes who are at different stages of their sporting careers.

## Supporting information

**S1 File.**
(XLSX)

**S2 File.**
(XLSX)

## Author Contributions

**Conceptualization:** Ryota Takamido, Yuji Yamamoto.

**Data curation:** Ryota Takamido.

**Formal analysis:** Ryota Takamido.

**Funding acquisition:** Ryota Takamido.

**Investigation:** Ryota Takamido, Keiko Yokoyama, Yuji Yamamoto.

**Methodology:** Ryota Takamido, Keiko Yokoyama, Hiroki Nakamoto, Yuji Yamamoto.

**Resources:** Ryota Takamido, Keiko Yokoyama.

**Software:** Ryota Takamido.

**Supervision:** Jun Ota, Yuji Yamamoto.

**Validation:** Hiroki Nakamoto.

**Visualization:** Ryota Takamido.

**Writing – original draft:** Ryota Takamido.

**Writing – review & editing:** Ryota Takamido.

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
