## [Decision Letter · Decision Letter 0]

19 Jun 2023

PONE-D-23-10689Interpersonal coordination analysis in bat-and-ball sports under a real-game situation: asymmetric interaction and delayed couplingPLOS ONE

Dear Dr. Takamido,

Thank you for submitting your manuscript to PLOS ONE. After careful consideration, we feel that it has merit but does not fully meet PLOS ONE’s publication criteria as it currently stands. Therefore, we invite you to submit a revised version of the manuscript that addresses the points raised during the review process.

We look forward to receiving your revised manuscript.

Kind regards,

Monika Błaszczyszyn

Academic Editor

PLOS ONE

Journal Requirements:

5. We note that Figures 1 and 2 in your submission contain copyrighted images. All PLOS content is published under the Creative Commons Attribution License (CC BY 4.0), which means that the manuscript, images, and Supporting Information files will be freely available online, and any third party is permitted to access, download, copy, distribute, and use these materials in any way, even commercially, with proper attribution. For more information, see our copyright guidelines: http://journals.plos.org/plosone/s/licenses-and-copyright.

a. You may seek permission from the original copyright holder of Figures 1 and 2 to publish the content specifically under the CC BY 4.0 license. 

Reviewers' comments:

Reviewer's Responses to Questions

**Comments to the Author**

1. Is the manuscript technically sound, and do the data support the conclusions?

Reviewer #1: Yes

Reviewer #2: Partly

2. Has the statistical analysis been performed appropriately and rigorously? 

Reviewer #1: Yes

Reviewer #2: Yes

3. Have the authors made all data underlying the findings in their manuscript fully available?

Reviewer #1: No

Reviewer #2: Yes

4. Is the manuscript presented in an intelligible fashion and written in standard English?

Reviewer #1: No

Reviewer #2: Yes

5. Review Comments to the Author

Reviewer #1: This paper aimed to analyze the structure of interpersonal coordination in bat-and-ball sports using a novel method (DTW) and provided interesting findings. The conclusions are scientifically supported by the data. Therefore, I recommend publication, but have some suggestions for improving the paper that the authors should consider.

L32-34 and others: The authors have listed two features - asymmetric interaction and delayed coupling - that are unique to bat-and-ball sports but can be found in other sports, such as the serve in tennis, badminton, and the penalty kick in soccer. If the author claims that these two features are different from these other sports, please provide the reasons why. If not, please reconsider the phrase "specific characteristics in ball sports."

L119: How many matches for which data was obtained?

L132 “optically synchronized”: What do you mean by "optically"?

L139 “national softball league”: This expression is a little ambiguous. Was it the highest league in the country? College league? Please specify.

L140 “more than nine balls in one game”: How did you determine this criterion?

L163 “normalized”: How did you normalize it? Z-scores?

L168 “time range” and time-curves in the figures: Authors indicate the time axes in the figures by the frame number; I understand the importance of the alignment indices in DTW analysis, but some readers (including me) might like to know in which time range these phenomena occur. For example, how about indicating frame and actual time together?

L176 “DTW”: In my opinion, the most important feature of this paper is the use of DTW for the analysis of pitcher-batter interactions. Overall, I would recommend emphasizing this point more. For example, add DTW as a keyword, mention it in the Introduction section, or provide a paragraph discussing the effectiveness of DTW in the Discussion section.

L292- & Fig. 7: Please discuss the details of the dependency. The authors found that the batter's motion was more unstable than the pitcher's. So, does a higher dependency mean that the batter responded to the pitcher's small fluctuations with a higher amplitude? Did a batter in a pair with a smaller R2 have a more stable motion that was not coupled with the pitcher's motion? Or was he/she unstable but not coupled with the pitcher? What type of pairs were those with a large R2?

L354 - 355: Please explain specifically the difference of "leader-follower" and "defender-attacker" relationships.

Throughout the paper: You should carefully check the sentence for correctness. e.g., L183-188 “and and”, “to to”, L236 “pitcheswere”, L256 & 258 “at ,”.

Reviewer #2: This study analyzed interpersonal coordination between pitchers and batters in bat-and-ball sports (softball) based on video in a real game and revealed two characteristics: asymmetric interaction and delayed coupling. While this topic is interesting and the results reported seem natural, I have several theoretical and methodological concerns, and would like to see more detailed analyses.

The first concern is the validity of using only head motion as the subject of the analysis. Although head motion and weight shift are used interchangeably in this paper (for example, l.100), the two are not always congruent. For example, batters are often instructed not to move their head back and forth too much as the weight shifts from the back leg to the front leg. Although back-and-forth head movement may still be observed, the relationship between head movement and weight shift may not be simple. If head movement is to be considered as weight shift, the rationale for this should be more elaborated.

The more essential issue is that the interaction between the pitcher and batter's motion is likely to manifest itself in areas other than head motion.

When pitchers throw a variety of pitches, they basically try to throw them with the same form as much as possible so that the batter cannot predict the pitch type (although sometimes they intentionally vary the timing of their pitches to confuse the batter). The reason why the pitcher's head motion is less variable is not because s/he can take charge, as the authors assume (l.63-66), but rather because it is more advantageous to throw as much as possible in the same way.

The batter, on the other hand, must deal with a variety of pitches, including those with widely different velocities. For this purpose, it is not advantageous to simply follow the movement of the pitcher, who is trying to reduce the clues as much as possible or dares to deceive the batter, but it is necessary to follow the behavior of the actual pitched ball. Even if the batter is influenced by the pitcher's movement, it is probably because the batter is actively changing the timing and form of hitting by sensing the subtle differences in form (not necessarily head movement) for each type of pitch. Experiments that simultaneously measured the whole-body motions of opposing pitchers and batters have shown that good batters can actually do this (Nasu et al., 2020). Even among top-level softball players, individual differences in prediction and adjustment ability are quite large. In the case of high-level batters, the greater variability in the batter's head motion may not be due to being caught up in the pitching form, but rather to adjusting to the ball. On the other hand, it is thought that low-level batters have a strong tendency to simply get caught up in the pitcher's movements. Therefore, even if a batter's motion is influenced by the pitcher's motion, it would at least be more useful to distinguish between simple entrainment and active adjustment. But can this be done by analyzing head movement alone? Active adjustment is often manifested in movements in areas other than the head. Also, while DTW and cross-correlation analyses can capture simple entrainment, it would be difficult to capture more complex adjustment by integration of various information.

In addition, the movements will be different when predictions are successful and when they are not. To examine this point, it is also necessary to know whether the hitting was successful or not. The authors consider this as one of the limitations (l.388), but is it possible to examine this point in the data already available?

l.106 "Hence, it can be assumed that, if the batter's movement is more affected by the pitching movement (Figure 1a), the weight-shift pattern of the batter is more unstable across different pitches than is that of the pitcher, and the changes in the batter's movements can be explained by changes in the pitcher's movement (Hypothesis 1). "

In light of the above, it is natural that the batter's weight shift battern (actually head movement pattern) is more variable than that of the pitcher, but the reason for this is not simply self-initiated or other-initiated. It is possible that the batter's fluctuating head movement pattern is not a result of being caught up in that of the pitcher, but rather a result of adjustments based on predictions comprehensively based on information such as the movement characteristics of the pitcher's other parts of the body and the ball's behavior.

l.110 "In addition, because batters must follow the pitcher's movement while predicting the pitched ball (Figure 1b), there is a consistent time delay between them that is close to the ball travel time (Hypothesis 2)."

It is natural that there is a certain (less than ball travel time) delay between the pitcher's and batter's motion, but it is also based on the sophisticated prediction described above. High-level batters dare to vary the timing of their swing with fastballs and changeups, but this is not simply a reflection of the timing of the pitcher's motion (Nasu et al., 2020). It would be necessary to analyze the data by pitch type, or at least by distinguishing between fast and slow pitches. Some excellent batters are able to adjust their timing for impact against a changeup by making adjustments midway through the motion, even if the timing at the start of the motion is that for a fastball. To get a hit, one does not necessarily have to hit the ball hard. Softball has a hitting technique called slap. Can head movement-only analysis adequately handle such diversity of hitting strategies?

l.357 "However, in contrast to Hypothesis 2, the mean time delay between the batter's and pitcher's movements was approximately 250 ms, which is smaller than the mean ball travel time of 410 ms. There are several potential reasons for this."

In addition to what the authors point out, another possibility is that the cues used by the batter in timing are motions of body parts other than the pitcher's head motion. Furthermore, the latency between the batter's acquisition of visual information, planning of the motion, and actual execution of the hitting motion would also need to be considered.

Minor points

l.133 from the side angle

The pitcher's image in Fig.2 does not appear to be from the side angle. Is this appropriate for analyzing the head motion in the front-back direction?

l.184 "to to"

l.187 "by and and , by"

l.188 "corresponds to" what?

l.252 equation missing （I assumed an ordinary cross-correlation function)

l.267 "shifted by" what?

6. PLOS authors have the option to publish the peer review history of their article (what does this mean?). If published, this will include your full peer review and any attached files.

Reviewer #1: No

Reviewer #2: No

---

## [Author Response · Author response to Decision Letter 0]

20 Jul 2023

Reviewer #1: 

This paper aimed to analyze the structure of interpersonal coordination in bat-and-ball sports using a novel method (DTW) and provided interesting findings. The conclusions are scientifically supported by the data. Therefore, I recommend publication, but have some suggestions for improving the paper that the authors should consider.

Response: Thank you for your positive appraisal of our manuscript. We have significantly modified the original manuscript on the basis of your comments. First, we added a discussion on DTW analysis to improve the interpretation of the results and clarify the effectiveness of this method. On the basis of the comments of Reviewer 2, we added a discussion on the effect of the prediction ability of the batter. Finally, our manuscript was rechecked by native English speakers to improve the sentences and correct grammatical errors throughout the manuscript. We believe that our manuscript has been made more appropriate by these corrections. The detailed answers to all of your questions and concerns are provided below in a point-by-point manner. 

Specific comments

1. L32-34 and others: The authors have listed two features - asymmetric interaction and delayed coupling - that are unique to bat-and-ball sports but can be found in other sports, such as the serve in tennis, badminton, and the penalty kick in soccer. If the author claims that these two features are different from these other sports, please provide the reasons why. If not, please reconsider the phrase "specific characteristics in ball sports."

Response: Thank you for your valuable feedback on our manuscript. As the reviewer indicated, two features of the interpersonal coordination of bat-and-ball sports (asymmetric interaction and delayed coupling) can be observed in other sports contexts. Hence, we admit that a revision of the phrase was necessary. From this perspective, the differences between bat-and-ball sports and other sports are in the frequency of the findings. Although two features can be observed in some parts of the entire game in other sports, they are more frequently observed and are the dominant factor in bat-and-ball sports. Hence, we believe that the word “dominant” is more appropriate than “specific.” On the basis of your comments, we have changed the word “specific” to “dominant” through the manuscript and added a new sentence (Lines 56–59).

2. L119: How many matches for which data was obtained?

Response: Thank you for this question. We used data collected from 16 official matches of the Japan women’s softball league. On the basis of the reviewer’s comment, we added a new sentence in the manuscript (Lines 138–139). 

3. L132 “optically synchronized”: What do you mean by "optically"?

Response: We apologize for any confusion that this term might have caused. In the data collection (recording) process, two video cameras were synchronized on the basis of the information of the light bulbs flashing in the same viewing field, and we represented this procedure as “optically synchronized.” We added a new sentence to the revised manuscript to clarify these procedures (Lines 144–146).

4. L139 “national softball league”: This expression is a little ambiguous. Was it the highest league in the country? College league? Please specify.

Response: We apologize for this vague information. The term “national softball league” refers to the “Japan women’s softball league (JSL),” which consists of company teams and has highest competition level in Japan; most players of the national softball team participate in this league. On the basis of the reviewer’s comment, we added an explanation for clarity (Lines 154–156). 

5. L140 “more than nine balls in one game”: How did you determine this criterion?

Response: Thank you for this question. The criteria are based on the following considerations: the batter is given a minimum of three pitches (three strike counts) for constructing the relationship between pitcher in an at-bat; three at-bats are given to a batter in a game; and nine balls (three pitches and three at bats) can be set as a criterion for the analysis of interpersonal coordination in real game contexts. On the basis of the reviewer’s comment, we added the sentences in the revised manuscript and clarified the criteria for readers (Lines 157–162).

6. L163 “normalized”: How did you normalize it? Z-scores?

Response: The term “normalization” in this paper means “Z-score normalization.” On the basis of the reviewer’s comment, we rephrased the word “normalized” to “normalized with Z-scores” (Lines 185–187).

7. L168 “time range” and time-curves in the figures: Authors indicate the time axes in the figures by the frame number; I understand the importance of the alignment indices in DTW analysis, but some readers (including me) might like to know in which time range these phenomena occur. For example, how about indicating frame and actual time together?

Response: We are grateful for your recommendations. On the basis of the reviewer’s comment, we modified Figs. 4 and 5 to include the actual time to provide more detailed information to readers.

8. L176 “DTW”: In my opinion, the most important feature of this paper is the use of DTW for the analysis of pitcher-batter interactions. Overall, I would recommend emphasizing this point more. For example, add DTW as a keyword, mention it in the Introduction section, or provide a paragraph discussing the effectiveness of DTW in the Discussion section.

Response: Thank you for your thoughtful critique and suggestions. We agree with the reviewer. Thus, we added a new paragraph in the Discussion section for the interpretation of the results of DTW and its effectiveness (Lines 378–395).

9. L292- & Fig. 7: Please discuss the details of the dependency. The authors found that the batter's motion was more unstable than the pitcher's. So, does a higher dependency mean that the batter responded to the pitcher's small fluctuations with a higher amplitude? Did a batter in a pair with a smaller R2 have a more stable motion that was not coupled with the pitcher's motion? Or was he/she unstable but not coupled with the pitcher? What type of pairs were those with a large R2?

Response: Thank you for these questions. The stability here represents the degree of consistency of the time series of the head velocity data when it is expanded and contracted along the time by DTW. When the change in pitching movement pattern is observed in some pitches (e.g., a relatively shorter weight-shift behavior than usual), the R2 value is increased if the batter’s hitting movement is also changed in accordance with it (e.g., also shows a relatively short weight-shift behavior). Therefore, the high R2 value represents that changes in the movement patterns of pitchers and hitters have similar tendencies (i.e., coupling). However, the largeness of the fluctuation is not considered in this analysis (the batter may respond with small fluctuations). 

Given that the batter also shows an R2 value > 0.90 (Fig. 6), there is a high possibility that a low R2 value indicates that a batter tends to perform stable hitting movements that are (relatively) not coupled with the pitcher’s motion. Therefore, there may be individual differences in the strategy of batters (strongly coupling with changeable movement vs. weakly coupling with stable movement), and these discussions bring us new insights of “what is the experts of interpersonal coordination.” On the basis of the reviewer’s comments and these discussions, we added a new paragraph to discuss these topics in the revised manuscript (Lines 378-395).

10. L354 - 355: Please explain specifically the difference of "leader-follower" and "defender-attacker" relationships.

Response: The difference between the “leader–follower” and “defender–attacker” relationships is mainly in the direction of the interaction. In a “leader–follower” relationship, the leader takes the initiative, and the other follows. In the “defender–attacker” relationship, the attacker tends to be in the dominant position, and both the attacker and defender need to observe their opponent’s action and change their movement. On the basis of the reviewer’s comment, we added an explanation of the meanings of both relationships (Lines 398–402).

11. Throughout the paper: You should carefully check the sentence for correctness. e.g., L183-188 “and and”, “to to”, L236 “pitcheswere”, L256 & 258 “at ,”.

Response: Thank you for your time and effort in reviewing our paper. We have rechecked the whole manuscript for errors. Our revised manuscript was sent to a native English speaker for proofreading.

Reviewer #2: 

This study analyzed interpersonal coordination between pitchers and batters in bat-and-ball sports (softball) based on video in a real game and revealed two characteristics: asymmetric interaction and delayed coupling. While this topic is interesting and the results reported seem natural, I have several theoretical and methodological concerns, and would like to see more detailed analyses:

Response: We thank the reviewer for your insightful comments and criticism. By taking both reviewers’ comments into account, we made some revisions to the original manuscript. The most significant change is the addition of a discussion on the effect of the prediction ability of the batter. As the reviewer indicated, there is a possibility that the prediction for the pitched ball affects the interaction between pitcher and batter; we cannot deny this perspective. Therefore, we need to sincerely present “what we can clarify from the analysis of this study” and “what we cannot clarify and need to investigate in the future” to the reader. Hence, we added two paragraphs to the Discussion section to clarify the above points. Furthermore, with respect to the further analysis mentioned by the reviewer (e.g., successful vs failure hitting), given that our dataset is unbalanced on the number of successful vs failure hitting or fast vs change-up pitch, it is difficult to perform a reliable analysis on these points even though we agree with the importance of the analysis from these aspects. Hence, we also mention the requirements of these analysis as the important future topic in the Discussion section. Finally, our manuscript was proofread by native English speakers to improve sentences and correct grammatical errors throughout the manuscript. We believe that our manuscript has been made more appropriate by these corrections. The detailed answers to all of the reviewer’s questions and concerns are provided below in a point-by-point manner. 

1. The first concern is the validity of using only head motion as the subject of the analysis. Although head motion and weight shift are used interchangeably in this paper (for example, l.100), the two are not always congruent. For example, batters are often instructed not to move their head back and forth too much as the weight shifts from the back leg to the front leg. Although back-and-forth head movement may still be observed, the relationship between head movement and weight shift may not be simple. If head movement is to be considered as weight shift, the rationale for this should be more elaborated.

Response: We appreciate the reviewer’s thoughtful critique and suggestions. As the reviewer indicated, head movement is not always congruent. We admit that the word “weight-shift pattern” may be misleading to the reader. Thus, we changed the word to “head movement pattern” in the revised manuscript.

However, we believe that there is a benefit in analyzing the interaction between pitchers and batters on the basis of the information of back-and-forth head movement. First, as shown in previous studies, the changes in pitching movement are reflected in the changes in a batter’s head movement (Takamido et al., 2022). Furthermore, these “global body positioning” behavior is more affected by advanced kinematic information in the preparation (coordination) phase before ball release, which is the main target of the current study (Müller and Abernethy, 2012). Even in skilled players, a pattern of back-and-forth head movement has been observed (Nakata et al., 2012). 

Although there is a possibility that the other body component information of the pitcher, such as the swinging of the arms, may be used for the interaction, the head movement of the pitcher and batter can serve as a clue for analyzing interpersonal coordination in bat-and-ball sports. On the basis of the reviewer’s comments and the above discussions, we added new sentences to explain the reason for selecting the head movement (Lines 103–114) in the Introduction section and the possibility of other types of interactions in the Discussion section (Lines 427–433).

Takamido R, Yokoyama K, Yamamoto Y. Hitting movement patterns organized by different pitching movement speeds as advanced kinematic information. Hum Mov Sci. 2022;81: 102908.

Müller S, Abernethy B. Expert anticipatory skill in striking sports: A review and a model. Res Q Exer Sport. 2012;83(2): 175-187.

Nakata H, Miura A, Yoshie M, Kudo K. Differences in the head movement during baseball batting between skilled players and novices. J Strength Cond Res. 2012;26(10): 2632-2640.

2. The more essential issue is that the interaction between the pitcher and batter's motion is likely to manifest itself in areas other than head motion. When pitchers throw a variety of pitches, they basically try to throw them with the same form as much as possible so that the batter cannot predict the pitch type (although sometimes they intentionally vary the timing of their pitches to confuse the batter). The reason why the pitcher's head motion is less variable is not because s/he can take charge, as the authors assume (l.63-66), but rather because it is more advantageous to throw as much as possible in the same way.

The batter, on the other hand, must deal with a variety of pitches, including those with widely different velocities. For this purpose, it is not advantageous to simply follow the movement of the pitcher, who is trying to reduce the clues as much as possible or dares to deceive the batter, but it is necessary to follow the behavior of the actual pitched ball. Even if the batter is influenced by the pitcher's movement, it is probably because the batter is actively changing the timing and form of hitting by sensing the subtle differences in form (not necessarily head movement) for each type of pitch. Experiments that simultaneously measured the whole-body motions of opposing pitchers and batters have shown that good batters can actually do this (Nasu et al., 2020). Even among top-level softball players, individual differences in prediction and adjustment ability are quite large. In the case of high-level batters, the greater variability in the batter's head motion may not be due to being caught up in the pitching form, but rather to adjusting to the ball. On the other hand, it is thought that low-level batters have a strong tendency to simply get caught up in the pitcher's movements. Therefore, even if a batter's motion is influenced by the pitcher's motion, it would at least be more useful to distinguish between simple entrainment and active adjustment. But can this be done by analyzing head movement alone? Active adjustment is often manifested in movements in areas other than the head. Also, while DTW and cross-correlation analyses can capture simple entrainment, it would be difficult to capture more complex adjustment by integration of various information.

Response: We would like to acknowledge the reviewer for their thorough review of our paper and for providing detailed feedback. As the reviewer indicated, there is a possibility that body component information other than head movement can be used for the interaction between the pitcher and batter, and it highly depends on the batter’s prediction ability. Furthermore, our analysis cannot discriminate passive entrainment from active adjustment and cannot capture more complex interaction. Hence, we added a new paragraph to discuss the possibility of these aspects and clearly show the limitation of our methodology to the reader (Lines 423–438).

Although there are some limitations in the analysis of this study, we consider that it is worthwhile to perform the analysis and obtain some new insights on the interaction between pitcher and batter. In this regard, we consider that the most important point is to clarify for the reader the contribution of our study: what we could clarify by the analysis of this study in the aspect of interpersonal coordination and what we cannot.

To clarify the contribution of our study to the reader, we added a discussion on the basis of a model of expert anticipation skills in striking sports (Müller and Abernethy, 2012) (Lines 439–455). From the model, during the hitting movement processes of the batter, there are mainly two types of adjustment behavior performed on the basis of visual information. The one is “global body positioning,” which is the adjustment of the position and pose of the entire body performed on the basis of early visual information such as a pitcher’s weight-shift movement. In this process, given that the batter does not have enough information for predicting the pitched ball accurately, the batter needs to couple with the pitching motion and to gather the information for knowing the future event. Furthermore, the other is the “precise spatio-temporal interception,” which adjusts the position and pose of the end-effector such as a bat to the throwing ball performed on the basis of the latter visual information, such as ball flight information. In this phase, the interaction is mainly affected by the prediction of the pitched ball trajectory and by the local body component, which contains information for knowing the type or location of the pitch, such as the throwing arm that can be used for interaction. 

Obviously, this study analyzed and focused on the former interaction. This study revealed the existence of the coordinate relationship between the pitcher and batter in real game contexts of bat-and-ball sports from the viewpoint of it. Hence, further investigations are required to analyze the latter type of interaction even though the former interaction should also be analyzed in more detail as the response to the next comment. We consider these discussions to be important for this paper.

3. In addition, the movements will be different when predictions are successful and when they are not. To examine this point, it is also necessary to know whether the hitting was successful or not. The authors consider this as one of the limitations (l.388), but is it possible to examine this point in the data already available?

Response: Thank you for this query. We agree that it is important to know and compare successful and unsuccessful hitting. Thus, we recorded the hitting results of the batters we analyzed in this study. However, given that the number of successful hits is quite small in real game situations, it is difficult to analyze the data from this aspect (in extreme cases, only two successful hits were observed in a game, and most batters has no successful data). It is also difficult to make a clear distinction between success and failure from 2D image information. Therefore, to verify this point, it is necessary to conduct further experiments on adjusting the difficulty of the task to obtain a sufficient amount of both success and failure data. From this aspects, a virtual reality system that can easily change the difficulty of hitting a ball can be a potential tool. On the basis of the comments and discussions, we added sentences in the revised manuscript (Lines 468–472).

4. l.106 "Hence, it can be assumed that, if the batter's movement is more affected by the pitching movement (Figure 1a), the weight-shift pattern of the batter is more unstable across different pitches than is that of the pitcher, and the changes in the batter's movements can be explained by changes in the pitcher's movement (Hypothesis 1). "

In light of the above, it is natural that the batter's weight shift battern (actually head movement pattern) is more variable than that of the pitcher, but the reason for this is not simply self-initiated or other-initiated. It is possible that the batter's fluctuating head movement pattern is not a result of being caught up in that of the pitcher, but rather a result of adjustments based on predictions comprehensively based on information such as the movement characteristics of the pitcher's other parts of the body and the ball's behavior.

Response: We are grateful for the reviewer's constructive suggestions (Comment No. 2), we agree that there is a possibility that the batter’s prediction ability affects the interaction between the pitcher and batter, and this point should be communicated to the reader and discussed in detail. This issue was also addressed in new paragraphs in Lines 423–455. 

5. l.110 "In addition, because batters must follow the pitcher's movement while predicting the pitched ball (Figure 1b), there is a consistent time delay between them that is close to the ball travel time (Hypothesis 2)."

It is natural that there is a certain (less than ball travel time) delay between the pitcher's and batter's motion, but it is also based on the sophisticated prediction described above. High-level batters dare to vary the timing of their swing with fastballs and changeups, but this is not simply a reflection of the timing of the pitcher's motion (Nasu et al., 2020). It would be necessary to analyze the data by pitch type, or at least by distinguishing between fast and slow pitches. Some excellent batters are able to adjust their timing for impact against a changeup by making adjustments midway through the motion, even if the timing at the start of the motion is that for a fastball. To get a hit, one does not necessarily have to hit the ball hard. Softball has a hitting technique called slap. Can head movement-only analysis adequately handle such diversity of hitting strategies?

Response: Thank you for your important comment. As the reviewer’s concern, we cannot catch-up such a detailed adjustment based on the predictions and variation of the hitting style of the batter. The target of this study is mostly the head movement behavior of the pitcher and batter coordinates with each other on a long-time scale from the pitcher's set position to the batter's swing. This study also aimed to obtain the big picture of the interpersonal coordination in bat-and-ball sports. Hence, those detailed techniques are investigated via the control experiment. This reviewer’s comment is also mentioned in new paragraphs in Lines 423–455. 

6. l.357 "However, in contrast to Hypothesis 2, the mean time delay between the batter's and pitcher's movements was approximately 250 ms, which is smaller than the mean ball travel time of 410 ms. There are several potential reasons for this."

In addition to what the authors point out, another possibility is that the cues used by the batter in timing are motions of body parts other than the pitcher's head motion. Furthermore, the latency between the batter's acquisition of visual information, planning of the motion, and actual execution of the hitting motion would also need to be considered.

Response: Thank you for your valuable feedback on our manuscript. As the reviewer indicated, there is a possibility that the batter couples with body parts other than the head of the pitcher or considers the required time for movement execution. On the basis of the reviewer's comment, we have added sentences in Lines 407–409 (movement executions time) and 420–422 (couple with other body parts).

Minor points

7. l.133 from the side angle

The pitcher's image in Fig.2 does not appear to be from the side angle. Is this appropriate for analyzing the head motion in the front-back direction?

Response: Thank you for this question. We basically recorded the image from the right side by using a camera set on the stand; however, owing to the limitation of the structure of the ballpark, we sometimes need to slightly change the view angle as the image. However, considering that the amount of head movement of the pitcher is large enough to discriminate the direction and given that we normalized the head movement data, we consider that it has acceptable accuracy for the analysis of this study. Owing to the ensures of the copy right, the image in Figure 2 has been changed to new image that was obtained using the same ballpark from the original version. On the basis of the comments from the reviewer, we added new sentences to improve clarity (Lines 149–153).

8. l.184 "to to", l.187 "by and and , by", l.188 "corresponds to" what?, l.252 equation missing (I assumed an ordinary cross-correlation function), l.267 "shifted by" what?

Response: Thank you for your comment. On the basis of the reviewer’s comments, we have rechecked the sentence for correctness throughout the paper. Our revised manuscript was sent to native English speakers for proofreading. We believe that these revisions and corrections have improved the paper.

We look forward to hearing from you and would be happy to make further changes if required.

---

## [Decision Letter · Decision Letter 1]

16 Aug 2023

Interpersonal coordination analysis in bat-and-ball sports under a real game situation: Asymmetric interaction and delayed coupling

PONE-D-23-10689R1

Dear Dr. Takamido,

We’re pleased to inform you that your manuscript has been judged scientifically suitable for publication and will be formally accepted for publication once it meets all outstanding technical requirements.

Kind regards,

Monika Błaszczyszyn

Academic Editor

PLOS ONE

Reviewers' comments:

Reviewer's Responses to Questions

**Comments to the Author**

1. If the authors have adequately addressed your comments raised in a previous round of review and you feel that this manuscript is now acceptable for publication, you may indicate that here to bypass the “Comments to the Author” section, enter your conflict of interest statement in the “Confidential to Editor” section, and submit your "Accept" recommendation.

Reviewer #1: All comments have been addressed

Reviewer #2: All comments have been addressed

2. Is the manuscript technically sound, and do the data support the conclusions?

Reviewer #1: Yes

Reviewer #2: Yes

3. Has the statistical analysis been performed appropriately and rigorously? 

Reviewer #1: Yes

Reviewer #2: Yes

4. Have the authors made all data underlying the findings in their manuscript fully available?

Reviewer #1: Yes

Reviewer #2: Yes

5. Is the manuscript presented in an intelligible fashion and written in standard English?

Reviewer #1: Yes

Reviewer #2: Yes

6. Review Comments to the Author

Reviewer #1: (No Response)

Reviewer #2: (No Response)

7. PLOS authors have the option to publish the peer review history of their article (what does this mean?). If published, this will include your full peer review and any attached files.

Reviewer #1: No

Reviewer #2: No

---

## [Editor Report · Acceptance letter]

21 Aug 2023

PONE-D-23-10689R1 

Interpersonal coordination analysis in bat-and-ball sports under a real game situation: Asymmetric interaction and delayed coupling 

Dear Dr. Takamido:

I'm pleased to inform you that your manuscript has been deemed suitable for publication in PLOS ONE. Congratulations! Your manuscript is now with our production department. 

Kind regards, 

on behalf of

Dr. Monika Błaszczyszyn 

Academic Editor

PLOS ONE